# Abundance Trends of Immature Stages of Ticks at Different Distances from Hiking Trails from a Natural Park in North-Western Italy

**DOI:** 10.3390/vetsci11100508

**Published:** 2024-10-15

**Authors:** Rachele Vada, Stefania Zanet, Elena Battisti, Ezio Ferroglio

**Affiliations:** Department of Veterinary Sciences, University of Turin, 10095 Grugliasco, Italy

**Keywords:** tick abundance, tick dispersal, wildlife–human interface, acarological risk, landscape management

## Abstract

**Simple Summary:**

Hiking trails in a natural park may increase the risk of tick exposure for people and pets, as ticks dropped by wildlife can easily latch onto individuals walking nearby. Understanding the range of movement of ticks dropped by wildlife hosts on trails, along with the acarological risks linked to park infrastructures, is crucial for guiding actions to reduce exposure to ticks and tick-borne pathogens. This study examines how tick abundance changes with increasing distance from hiking trails, focusing on a range of 0 to 4 m. We observed a decreasing trend in tick abundance that varies by developmental stage: nymphs show a steady decline in abundance as distance increases, while larvae exhibit a peak near the trail, followed by a sharp drop at greater distances. These findings highlight how the immediate proximity of a hiking trail may serve as a potential source of tick bites for both humans and pets.

**Abstract:**

Hiking trails may act as hotspots at the wildlife–human interface, posing an acarological risk for people and their pets. Ticks that are maintained in the environment by wild animals may quest on people walking along the trails. Assessing the risk of tick bites for people involved in outdoor activities is a further step in mitigating the risk of tick-borne diseases. This work describes the variation of tick abundance along a gradient of distances from hiking trails, where wildlife passage is favored by higher accessibility. Hiking trails with dense vegetation on the sides were sampled for ticks along a 100 m dragging transect, located in a natural park in North-Western Italy. Additional transects were replicated at 1, 2 and 4 m away from the trail on both sides. After morphological identification, descriptive statistics and modeling were applied to determine the abundance patterns across distances. Larvae were most abundant near the trail, peaking at 1 m and dropping sharply at further distances. Nymphs showed a more gradual and consistent decrease at progressing distance from the trail. Few adults were collected, preventing the identification of a clear trend. With higher tick abundance, the immediate vicinity of hiking trails may represent a source of acarological risk for humans and pets.

## 1. Introduction

Peri-urban parks are characterized by a high frequency of direct and indirect contact among wildlife, domestic animals and humans, which can boost the risk of transmission of zoonotic or epizootic pathogens [1]. Hiking trails, especially at nighttime, are frequently used by several wild mammals [2]; prey species (from ungulates to small mammals) seem in some cases to be attracted by busier trails, presumably using humans as a deterrent against predators activity [3], whereas mesocarnivores can use trails for more efficient hunting and traveling [4]. However, the extent of such a behavior may vary across species and the avoidance of human tracks has been registered, especially for deer species [5,6] and some rodent species [7]. As shared environments, these trails may serve as preferential sites for the transmission of parasites and zoonotic pathogens, including tick-borne zoonoses through the risk of tick bites.

Several studies have tried to characterize how recreational areas in natural parks expose people and their pets to acarological risk. This risk may be associated with landscape and trail characteristics [8,9], but also with visitors’ behavior [10], in addition to elements such as vegetation and host presence [11,12]. Hiking trails have also been described as hotspots for tick passage from wild to human hosts [11,13,14]. Footpaths in protected areas of Northern Italy have been sampled, confirming both the presence of the acarological risk and the zoonotic risk associated with tick-borne diseases [15,16,17].

When not on the host, ticks are subject to a certain range of dispersion. A study in Poland and Slovakia suggested that ticks tend to be more abundant on animal trails than at a range of 5 m distance from the trail [13], but there is limited literature available on the topic. While many studies focus on the abiotic factors that influence *Ixodes ricinus* locomotor activity, such as temperature, humidity, electromagnetic fields and light/darkness hours [18,19,20], dispersal models have been provided for *Ixodes scapularis* [21] and have been studied in other species such as *Dermacentor reticulatus* [22]. For *I. ricinus*, the maximum distance of attraction with carbon dioxide traps has been estimated to be up to 3.5 m for adult females and 1 m for nymphs [23]. Information on dispersion behavior would not only broaden the knowledge about ticks’ ecology but would also represent a significant factor to be considered in risk assessments for the transmission of vector-borne diseases (i.e., Lyme disease, babesiosis and tick-borne encephalitis). Additionally, it may inform infrastructure design and management strategies in peri-urban parks in order to reduce the risk.

This study aims to determine the gradient of environmental ticks’ distribution at different distances from pathways, an issue that has still been little explored despite its relevance to addressing public recommendations to reduce human exposure to ticks.

## 2. Materials and Methods

### 2.1. Study Area

The study was performed in La Mandria Regional Park in Piemonte, Italy. It is a fenced, protected area of 6571 Ha at an average altitude of 386 m above sea level. Due to the absence of an altitude gradient and a small area, the vegetation type is homogeneous in the whole area, with a total of 51.4% of the park covered in deciduous forest (made mainly of oak, hornbeam and ashwood) and 30.2% covered in grassland for mowing fields (Corine Land Cover data, downloaded from https://geoportale.igr.piemonte.it/cms/, accessed on 30 August 2024). The annual precipitation in 2020 was 898.2 mm and it registered an maximum temperature of 32 °C and a minimum temperature of −4 °C). It welcomes an average of 2000 visitors every day. Ungulate culling for population control is constant throughout the year. Few horse farms are present, but no livestock is kept in the park and dogs are not allowed, except for the ones owned by farmers. Regarding the wildlife population, ungulates are present at high densities [24]. Ticks and tick-borne pathogens that may spill over to humans have been recorded with a high prevalence in the study area; for instance, a prevalence of almost 45% of *Babesia* spp. has been recorded in red deer by Zanet and colleagues [25]. The mammal species of larger sizes that are most abundant in the park include wild boar (*Sus scrofa*), roe deer (*Capreolus capreolus*), red deer (*Cervus elaphus*) and fallow deer (*Dama dama*), as well as wolves (*Canis lupus*) and mesocarnivores such as foxes (*Vulpes vulpes*) and badgers (*Meles meles*) [24,26]. The abundance of other mustelids such as the European pine marten (*Martes martes*) and the beech marten (*Martes foina*) is comparatively low.

### 2.2. Sampling Design

We selected three sampling sites, constituting a main trail with dense and homogeneous grass or undergrowth (20–30 cm height) on the sides. Such vegetation would constrain the passage of larger size mammals (wild ungulates, wolves, foxes and badgers) to the trail, while rodents, other mustelids and reptiles would have easier access to the vegetation, in addition to different behaviors in relation to trail use. To maximize the possibility of collecting ticks, all sampling sites were identified in deciduous forest habitats. Additionally, no sign of passage of mentioned wild species, such as scats, tracks or vegetation damage [27] was detected outside of the trail, while the trails presented signs of passage. At each sampling site, we conducted dragging/flagging linear transects of 100 m in length, parallel to the trail. These transects were performed at perpendicular distances of 0 m, 1 m, 2 m and 4 m from the trail on both sides (Figure 1). A 1 square meter white cloth was used for sampling and was checked every 20 m. To account for seasonal differences in tick abundance and biases related to sampling events, dragging/flagging was repeated at each transect every two weeks from May to August 2021, for a total of 9 repetitions. All of the sites were sampled on the same day, avoiding the central hours during summer. Days with significant rain events, as well as the 48 h immediately following these, were excluded. Ticks were separately collected for each transect and stored in 70% ethanol for subsequent morphological identification [28,29,30].

In the proximity of each sampling site, we placed a sensor (Temperature and Humidity Data Logger PRO-USB-2, RS, Corby, UK) set to register the temperature and humidity hourly, 15 cm above the ground. The saturation deficit was calculated per day [31,32] and an average value for the period between the repetitions was retrieved.

### 2.3. Statistical Analysis

Statistical analysis was performed in R studio [33]. We implemented descriptive statistics to highlight tick distribution at different distances from the trail. We plotted the average abundance of ticks per transect, encompassing all developmental stages and species and larvae and nymphs separately (adults were disregarded due to poor representation). To explore different behaviors among species, we also plotted the two most abundant genera recorded in our study, namely *Ixodes* and *Haemaphysalis*. Both species are of interest as vectors of zoonotic diseases [34]. 

After explorative analysis, we described trends in tick abundance by structuring Generalized Additive Models [35], an implementation of General Linear Models that allows us to describe non-linear relationships with the explanatory variables (called smooth terms). This aspect is particularly fitted for ecological modeling and, in our case, to describe the trends in ticks’ abundance, which we expected to not be linear. We generated models for the total number of ticks, nymphs and larvae, implementing date and average saturation deficit as linear predictors and the distance from the hiking trail as the smooth parameter. The sampling site was tested as a weight factor. The models were selected based on the deviance explained, the significance of coefficients and the residual analysis.

## 3. Results

A total of 181 ticks were collected, for an average abundance of 0.01 tick/m^2^; these ticks comprised 132 larvae, 46 nymphs and 3 adults (Table 1). 

The majority (109) were identified as *I. ricinus*, followed by *Haemaphysalis punctata* (63), *Rhipicephalus sanguineus* complex (5), *Haemaphysalis concinna* (3) and *Dermacentor* sp. (1). Tick abundance is shown to increase over time, except for the months of June and August (Table 2).

Tick abundance was not constant throughout all the repetitions for the three sampling points, with a general increasing trend culminating in peaks during summer and an additional peak in May for sampling site B. However, nymphs alone present a peak earlier in time (end of the spring), while larvae present a peak during summer (Figure 2).

The trend in tick abundance at a progressive distance from trails differed by developmental stage (Figure 3). Most larvae were collected within 1 m, with the average abundance increasing by 59.2% from 0 m to 1 m. We registered the lowest average abundance at 2 m (decrease of 72.2% from 1 m to 2 m) and a final increase at 4 m (69.6% from 2 m to 4 m). The average abundance at 4 m was still 26.2% lower than at 1 m. This trend was also visible for the total number of ticks and *Haemaphysalis* spp.

The average abundance of nymphs decreased when becoming further from the trail: −43.6% from 0 m to 1 m, −6.8% from 1 m to 2 m and −73.1% from 2 m to 4 m and showing a very limited presence at 4 m. This trend was also identified for *I*. *ricinus.*

Due to the limited number of specimens collected, we did not identify abundance trends for adults. Despite the difference in average abundance, a similar trend was shown at all sampling sites (Table 3).

Based on the deviance explained, the significance of coefficients and residuals performance, we identified three models describing, respectively, the abundance of the total number of ticks (deviance explained 22.5%, Poisson distribution), larval abundance (deviance explained 35.9%, Poisson distribution) and nymphal abundance (deviance explained 35.9%, negative binomial distribution). All models introduced the date and saturation deficit as linear predictors and the distance from the trail as a smooth term. Date has a significant positive coefficient (*p* < 0.05) in the general and the larval model and a significant negative coefficient in the nymphal model, while saturation deficit has a significant positive coefficient (*p* < 0.05) in all models (Table 4).

Smooth terms are significant (*p* < 0.05) in all three models, showing trends that remark the ones highlighted by descriptive statistics (Figure 4), with a peak at 1 m for the general model and the larval model and a progressive decrease for the nymphs model.

## 4. Discussion

Hiking trails can be appealing for wildlife movement, particularly when dense vegetation along the sides obstructs passage, making the trails the easiest route to navigate. Our data showed a higher abundance of ticks at 0 m and 1 m distance with a heavy drop at 2 m, which is in line with the results previously obtained [13], suggesting that ticks’ abundance decreases when moving away from the path of animals or humans. The proximity of a pathway is a convenient spot for questing ticks to find a host, as the probability of passage is higher, and the adjacent vegetation creates a more suitable microhabitat for the tick’s activity and survival. 

The average ticks’ abundance observed in our study was lower than those reported in other studies conducted at higher latitudes. For example, Van Gestel and colleagues [14] recorded a density of nymphs of 0.46 per square meter, while Mols and colleagues a density of 0.75 per square meter [6]. However, studies conducted in nearby areas reported a lower density, though this was still higher than what we observed [17]. Differences in habitat characteristics, such as temperature, humidity and vegetation type, do play a role in influencing ticks’ abundance. Additionally, Hasset and colleagues [10] noted that tick density was greater in areas with less human presence [12]. This could also be observed when referring to a study [36] undertaken concurrently with the present one in the forest and grassland parcels of the park, where the registered abundance was higher, but sampling was also performed deeper in the forest/meadows.

While the total abundance of ticks per sampling site and repetition varied significantly between sites, this variation did not interfere with identifying a general trend. Factors such as the movement rates of different species and the types of vegetation present may account for these differences. A lower presence of wild ungulates and mesocarnivores was indeed correlated with a lower ticks abundance in the park [36].

Nymphs showed a decreasing pattern when collections were gradually taken further away from the trail and they were rarely collected at a 4 m distance, corroborating the thesis of a limited dispersion previously hypothesized with carbon dioxide traps [23]. Similarly, larvae were also concentrated along the proximity of the trail. Besides dispersion behavior, and consistent with the hypothesis that larvae have limited or no movement in the environment [37], the accessibility of the host to the area on the sides of the trail likely played a role in shaping the observed patterns. Small mammals, which contribute to the presence of questing nymphs in the environment, have better access through dense vegetation, and some species may also show avoidance behaviors towards trails [7], thus maintaining nymphal abundance more consistently even further away.

The pattern shown by different species was biased by the composition in terms of developmental stages: *Haemaphysalis* spp., 91% of which was constituted by larvae, followed the larval pattern, while *I. ricinus*, with a more even ratio of larvae–nymphs (56–41%, respectively), followed a more uniform decrescent pattern. Similar considerations can be addressed to the trend for the total average abundance of ticks. Therefore, our results could not allow us to draw conclusions about different dispersal behavior among tick species. Trends were identified through the Generalized Additive Models: the complexity of tick ecology may be the reason for the relatively low deviance explained, but the pattern described with the smooth terms clearly represents the trend observed with descriptive statistics.

The three models also allow us to remark on the effect of the progression of the season on ticks abundance, which is additive for larval stages and detractive for nymphs, whose population peak is usually registered between the end of spring and the beginning of summer [38,39]. The saturation deficit showed a significant impact on ticks activity; although a positive effect on ticks activity was registered only by Tagliapietra and colleagues [32], on most occasions, it has been detected as a limiting factor [31].

The composition, in terms of species and developmental stages, of the sampled ticks’ population reflected the literature data [34], with a higher abundance of *I. ricinus* and larvae, and a decreasing proportion of, in order, nymphs and adults. The sampling season may primarily account for the low number of adults collected [38], in addition to them being generally less abundant than in immature stages. Nymphs are consistently less abundant than larvae at all distances, despite questing on the upper stratum of the vegetation [40], which makes them more easily captured by the dragging/flagging cloth, suggesting that our sampling strategy was effective in also detecting the stages that usually quest closer to the ground.

Several studies in the European context have related park infrastructures to the risk of tick bite [11,14]; despite remarking that tick abundance is higher in the forest interior or in areas where fewer people have access, acarological risk is still relevant along trails and, with that, the risk for tick-borne zoonoses [8,13,14,17]. Animal presence, in addition to vegetation characteristics, is the fundamental driver for such exposure [6,12]. Additionally, visitors’ behavior contributes to defining the risk for people engaging in outdoor activities [10,11]. For instance, humans and their pets are more likely to spend more time in low vegetation spaces or along trails than in more remote areas. A study in the USA showed that, despite knowing of the risk of a tick bite, people believed they had a minimal risk of tick encounters and consequently tended not to conduct tick checks [10].

## 5. Conclusions

Analyzing the risk of tick-borne pathogen transmission in a park demands a focused consideration of infrastructures such as hiking trails. Developing integrated management strategies of infrastructures necessitates evaluating not only their impact on wildlife welfare and behavior but also their potential for drawing ticks closer to people. Our findings show that hiking trails exhibit higher tick abundance, increasing the acarological risk to both passing visitors and their pets.

## Figures and Tables

**Figure 1 vetsci-11-00508-f001:**
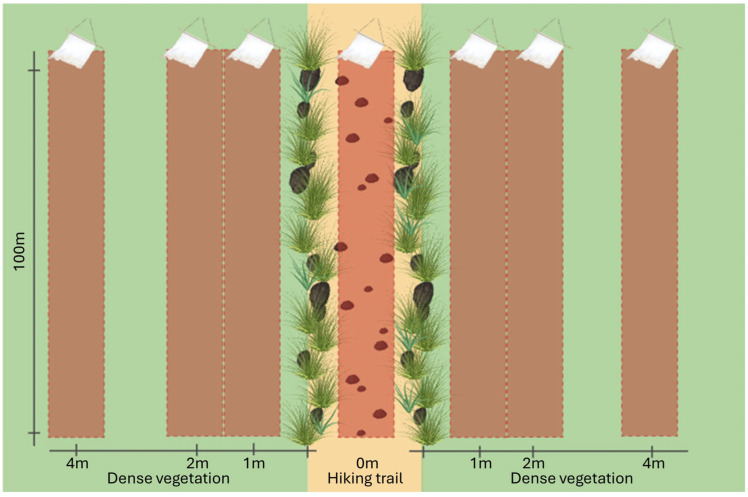
Transect deployment at each sampling site. Differentiating the hiking trail (light yellow) from the dense vegetation (light green), the dragging transects are plotted on the map (the brown rectangles represent a surface of 1 m width and 100 m length). Their distances are the following: one transect is on the trail (0 m) there is one in the vegetation for each side at 1 m, 2 m and 4 m.

**Figure 2 vetsci-11-00508-f002:**
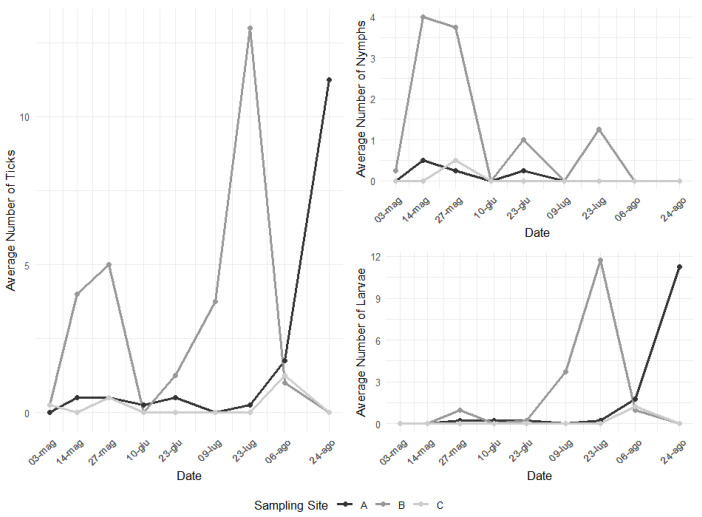
Fluctuation of tick abundance across the 9 repetitions in the three sampling sites.

**Figure 3 vetsci-11-00508-f003:**
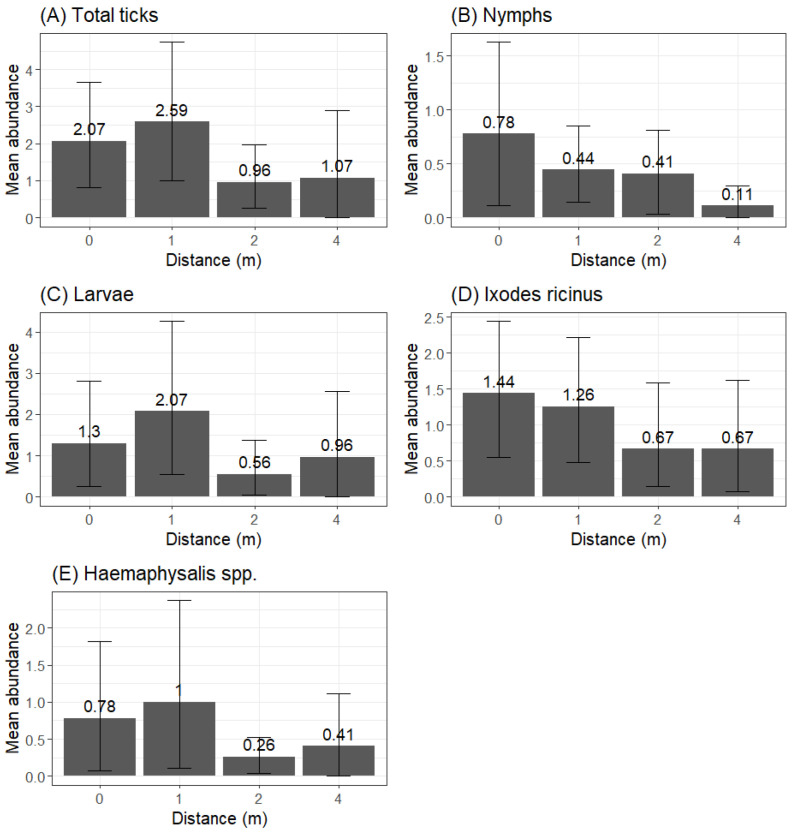
Plot illustrating the distribution of the average number of ticks at different distances (0 m, 1 m, 2 m and 4 m) from human trails for the following: (**A**) total amount of ticks, (**B**) nymphs, (**C**) larvae, (**D**) *Ixodes ricinus* species and (**E**) *Haemaphysalis* spp. species.

**Figure 4 vetsci-11-00508-f004:**
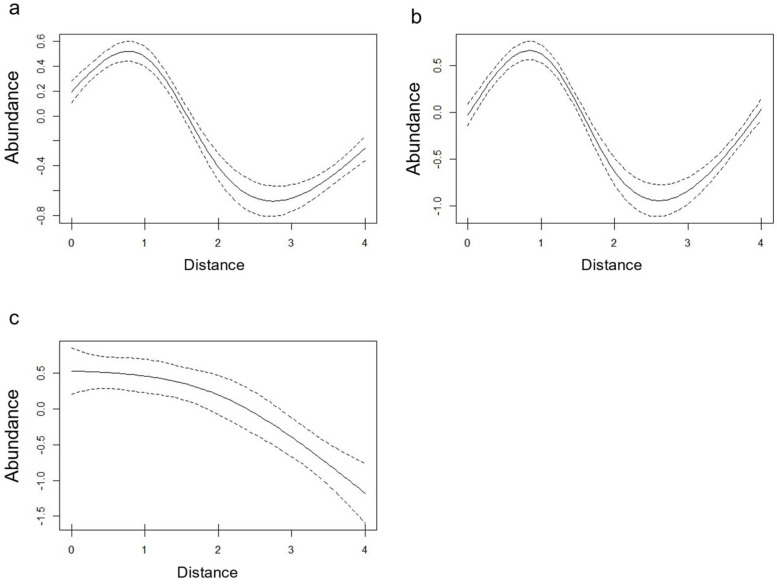
Plot of smooth term (distance from hiking trail, *x* axis) effects on tick abundance (*y* axis). Effects are plotted for the general model (**a**), the larval model (**b**) and the nymphal model (**C**).

**Table 1 vetsci-11-00508-t001:** Total count of ticks for species and developmental stage.

	Adults	Nymphs	Larvae	Total
*Dermacentor* sp.			1	1
*Haemaphysalis concinna*		3		3
*Haemaphysalis punctata*	1	2	60	63
*Ixodes ricinus*	2	41	66	109
*Rhipicephalus sanguineus* complex			5	5
Total	3	46	132	181

**Table 2 vetsci-11-00508-t002:** Average number of collected ticks for each species by month.

	*I. ricinus*	*H. punctata*	*H. concinna*	*Dermacentor* sp.	*Rhipicephalus sanguineus* Complex	Total
May	1.06	0.11	0.06	0.00	0.00	1.23
June	0.21	0.08	0.04	0.00	0.00	0.33
July	1.67	1.04	0.00	0.00	0.13	2.84
August	1.08	1.33	0.00	0.04	0.08	2.53

**Table 3 vetsci-11-00508-t003:** Average number of collected ticks (considering together all species and developmental stages) by sampling site at each distance.

	Sampling Site A	Sampling Site B	Sampling Site C
0 m	2.89	3.11	0.22
1 m	2.89	4.22	0.67
2 m	0.56	2.33	0.00
4 m	0.33	2.89	0.00
Total	1.67	3.14	0.22

**Table 4 vetsci-11-00508-t004:** Parametric coefficients and approximate significance of smooth terms for each model.

General			
Parametric Coefficients	Estimate	Std. Error	Pr (>|z|)
(Intercept)	−2.50	0.12	<2 × 10^−16^
Date	0.07	0.01	1.43 × 10^−9^
Saturation deficit	0.7	0.03	<2 × 10^−16^
**Smooth terms**	**edf**	**Ref.df**	***p*-value**
s(distance)	3	3	<2 × 10^−16^
**Larvae**			
**Parametric Coefficients**	**Estimate**	**Std. Error**	**Pr (>|z|)**
(Intercept)	−4.42	0.15	<2 × 10^−16^
Date	0.34	0.02	<2 × 10^−16^
Saturation deficit	0.65	0.03	<2 × 10^−16^
**Smooth terms**	**edf**	**Ref.df**	***p*-value**
s(distance)	3	3	<2 × 10^−16^
**Nymphs**			
**Parametric Coefficients**	**Estimate**	**Std. Error**	**Pr (>|z|)**
(Intercept)	−3.45	0.51	1.95 × 10^−11^
Date	−0.62	0.07	<2 × 10^−16^
Saturation deficit	1.41	0.19	1.34 × 10^−13^
**Smooth terms**	**edf**	**Ref.df**	***p*-value**
s(distance)	2.06	2.4	2.25 × 10^−7^

## Data Availability

Data will be made available upon reasonable request to the authors.

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
