# Peer review of "Abundance Trends of Immature Stages of Ticks at Different Distances from Hiking Trails from a Natural Park in North-Western Italy"

_vetsci, 2024, doi:10.3390/vetsci11100508_

Round 1
Reviewer 1 Report
Comments and Suggestions for Authors
Dear Authors,
I have appreciated the article “Where the host steps, the tick thrives: description of abundance trends of ticks sampled at different distances from hiking trails”
Regarding the realization of this study I don't have major comments.
There are only a few suggestions.
Best regards
Line number:
Material & Methods:
Line 54 -58: “Regarding wildlife population, La Mandria is……colleagues[10] – Not clear. Change the sentence.
Results:
Line 103 – Tick abundance is shown to increase over time – Except for the month of June and August.
Line 116 – Ixodes ricinus and Haemaphysalis punctata – no italics
Line 125 – change “per” with “by”
Discussion
Line 140 – 142: “In addition…patterns” – Not clear. Change the sentence.
Line 167 – months”…add with the greatest numbers in july.
References
Line 203; 216;218;219 no italics
Author Response
We thank the reviewer for their comments and suggestion, which we incorporated in the manuscript thus improving the overall quality of it.
Material & Methods
Comment: Line 54 -58: “Regarding wildlife population, La Mandria is……colleagues[10] – Not clear. Change the sentence.
Answer: the sentence was split to improve clarity
Results
Comment: Line 103 – Tick abundance is shown to increase over time – Except for the month of June and August.
Answer: the sentence was corrected according to the suggestion
Comment: Line 116 – Ixodes ricinus and Haemaphysalis punctata – no italics
Answer: we corrected the italics
Comment: Line 125 – change “per” with “by”
Answer: corrected
Discussion
Comment: Line 140 – 142: “In addition…patterns” – Not clear. Change the sentence.
Answer: sentence was rephrased to improve clarity
Comment: Line 167 – months”…add with the greatest numbers in july.
Answer: sentence was modified accordingly
References
Comment: Line 203; 216;218;219 no italics
Answer: italics was corrected through the whole reference list
Reviewer 2 Report
Comments and Suggestions for Authors
See attached file.

Certain terms and verbage require editing to improve English.
Author Response
We thank the reviewer for their comments and suggestion, which we incorporated in the manuscript thus improving the overall quality of it.
Comment: it would help to explain a bit more about the relative difference in mobility of larvae and nymphs, as well as the host differences that influence (either way) where they would be more abundant relative to distance from the trail this would aid the explanation about those differences
Answer: we have implemented the explanation about ticks locomotor activity (lines 54-57). However, not much information is available on the topic, as many studies have targeted different tick species to the ones collected in the present study (I. ricinus), which makes results comparison quite hard. We corrected the paragraph mentioning host differences that would influence tick distribution on the sides of the path (lines 219-228).
Comment: The number of ticks collected appears to be low overall – only 0.017 ticks/square meter; comparison with other studies would benefit the reader.
Answer: we added a paragraph in the discussion (lines 202-212).
Comment: it is important to identify the number of transects that had zero ticks and whether abundance was related to time (the 9 reps across 18 weeks). This might prove useful for design of future studies and better describe risk
Answer: we added a graph showing ticks abundance across time (line 153) and we commented the general trend in tick abundance per site with the progression in time (lines 241-244).
Comment: Area C was nearly devoid of ticks relative to A and B (table 3) – this needs to be discussed because the 3 areas were assumed similar at the onset. Further, those zeros in C have measurable influence on averages and interpretation. It might be more valuable to focus analysis on A and B?
Answer: Following reviewer’s suggestions, we tried to exclude from the analysis point C, however it did not improve the results in terms of significance, coefficients or deviance explained, so eventually we left the point in the analysis. Also, we checked for zero inflation in the models, but this assumption was not matched and we kept the poisson/negative binomial distribution.
Comment: Tick abundance in areas A and B was not too similar relative to overall abundance or distance (Table 3). It is interesting to consider the variation among A, B, C given the controls in sampling – why was B such a hotspot and C the opposite?
Answer: this aspect was commented in the discussion (lines 213-218).
Reviewer 3 Report
Comments and Suggestions for Authors
Revise the title and the abstract to reflect that this work was done in Italy.
Line 6: May quest "for" and add "people or their pets"
Line 14 "nymph"
Line 16 tick abundance
Line 24 What species?
Line 24-28. I can't follow the flow of this section. Please rewrite.
Line 29 locate "prey"
LIne 29-30. Rewrite.
Line 30: As shared environments, those trails
Also at this point you have talked about pathogens but not really ticks. You can't have indirect transmission of ticks. That is direct acquisition.
Line 32 What does this mean? After descending the host
Line 35: Such dispersal is likely to be directed towards suitable micro habitats for ticks’ survival and quest, such as shrubs. -Do you have a reference for this?
Suggest adding some more park risk or human risk in wild areas info
Padgett KA, Bonilla DL. Novel exposure sites for nymphal Ixodes pacificus within picnic areas. Ticks Tick Borne Dis. 2011 Dec;2(4):191-5. doi: 10.1016/j.ttbdis.2011.07.002. Epub 2011 Sep 25. PMID: 22108011.
Hahn MB, Feirer S, Monaghan AJ, Lane RS, Eisen RJ, Padgett KA, Kelly M. Modeling future climate suitability for the western blacklegged tick, Ixodes pacificus, in California with an emphasis on land access and ownership. Ticks Tick Borne Dis. 2021 Sep;12(5):101789. doi: 10.1016/j.ttbdis.2021.101789. Epub 2021 Jul 13. PMID: 34280699; PMCID: PMC9379859.
Wierzbicka A, RÄ…czka G, Skorupski M, Michalik J, Lane RS. Human behaviors elevating the risk of exposure to Ixodes ricinus larvae and nymphs in two types of lowland coniferous forests in west-central Poland. Ticks Tick Borne Dis. 2016 Oct;7(6):1180-1185. doi: 10.1016/j.ttbdis.2016.07.018. Epub 2016 Aug 1. PMID: 27499188.
Those are just a few. You need to go back and do a more thorough literature search. Ticks and trails as keywords together on pubmed yields a ton of pertinent information.
Line 43-45. Maybe it hasn't been explored in Italy? It has been explored elsewhere (hence my previous notes)
Line 48 What is a.s.l.?
Line 50. Usually the main type of vegetative cover is listed here
Line 53 Which ungulates? Are there livestock? Are dogs allowed? off leash?
Line 54-55 In regard the the wildlife population, ...high ungulate density
Line 57 recorded. For instance,
Line 59 what ungulates
Line 70-ok so did you see these signs on the trail?
Line 74- checking the flag every 100 m is a bit too long. You could have potentially lost ticks
There probably needs to be some discussion about how removal of ticks does/doesn't affect the tick population there
Line 70- "broad" hiking trails. I did not see mention before of how wide the trails were and why it matters? That is actually an interesting question.
LIne 135 Nymphs showed a decrease in number getting further from the trail (insert statistical significance) Same for the rest of the discussion. Insert statistical significance along with the descriptive statistics.
Line 137 Carbon dioxide traps
Line 138 139 decreased by
Line 146 further away
Line 173 "findings show" and "risk to"
Comments on the Quality of English Language
This article could use a review by a native English speaker. Many of the questions I have with the text could just be a translation issue.
Author Response
We thank the reviewer for their comments and suggestion, which we incorporated in the manuscript thus improving the overall quality of it.
Comment: Revise the title and the abstract to reflect that this work was done in Italy.
Answer: We added the location reference (Natural Park in North-Western Italy) in title and abstract as requested
Comment: Line 6: May quest "for" and add "people or their pets"
Answer: the sentence was modified accordingly.
Comment: Line 14 "nymph"
Answer: corrected.
Comment: Line 16 tick abundance
Answer: corrected.
Comment: Line 24 What species?
Answer: we have specified “wild mammals”, which would include all ungulate, carnivorous and small mammal species.
Comment: Line 24-28. I can't follow the flow of this section. Please rewrite.
Answer: sentence was rephrased to improve clarity
Comment: Line 29 locate "prey"
Answer: we specified what we mean as for “prey species”
Comment: Line 29-30. Rewrite.
Answer: sentence was rephrased to improve clarity.
Comment: Line 30: As shared environments, those trails
Answer: corrected.
Comment: Also at this point you have talked about pathogens but not really ticks. You can't have indirect transmission of ticks. That is direct acquisition.
Answer: We have clarified the statement.
Comment: Line 32 What does this mean? After descending the host
Answer: Substituted by “detached”
Comment: Line 35: Such dispersal is likely to be directed towards suitable micro habitats for ticks’ survival and quest, such as shrubs. -Do you have a reference for this?
Answer: This sentence was removed.
Comment: Suggest adding some more park risk or human risk in wild areas info
Padgett KA, Bonilla DL. Novel exposure sites for nymphal Ixodes pacificus within picnic areas. Ticks Tick Borne Dis. 2011 Dec;2(4):191-5. doi: 10.1016/j.ttbdis.2011.07.002. Epub 2011 Sep 25. PMID: 22108011.
Hahn MB, Feirer S, Monaghan AJ, Lane RS, Eisen RJ, Padgett KA, Kelly M. Modeling future climate suitability for the western blacklegged tick, Ixodes pacificus, in California with an emphasis on land access and ownership. Ticks Tick Borne Dis. 2021 Sep;12(5):101789. doi: 10.1016/j.ttbdis.2021.101789. Epub 2021 Jul 13. PMID: 34280699; PMCID: PMC9379859.
Wierzbicka A, RÄ…czka G, Skorupski M, Michalik J, Lane RS. Human behaviors elevating the risk of exposure to Ixodes ricinus larvae and nymphs in two types of lowland coniferous forests in west-central Poland. Ticks Tick Borne Dis. 2016 Oct;7(6):1180-1185. doi: 10.1016/j.ttbdis.2016.07.018. Epub 2016 Aug 1. PMID: 27499188.
Those are just a few. You need to go back and do a more thorough literature search. Ticks and trails as keywords together on pubmed yields a ton of pertinent information.
Answer: we have inserted a paragraph in the introduction and implemented discussion with suggested and more citations (lines 44-50 and 257-266).
Comment: Line 43-45. Maybe it hasn't been explored in Italy? It has been explored elsewhere (hence my previous notes)
Answer: we have inserted a paragraph in the introduction and implemented discussion with examples of works in Italy as well (i.e. Millet et al., 2019).
Comment: Line 48 What is a.s.l.?
Answer: above sea level, we have extended the abbreviation.
Comment: Line 50. Usually the main type of vegetative cover is listed here
Answer: we have integrated a clarification of the main vegetation species in the park (line 71).
Comment: Line 53 Which ungulates? Are there livestock? Are dogs allowed? off leash?
Answer: we have included all of those details in the description (lines 77-78).
Comment: Line 54-55 In regard the the wildlife population, ...high ungulate density
Answer: corrected.
Comment: Line 57 recorded. For instance,
Answer: corrected.
Comment: Line 59 what ungulates
Answer: we clarified the sentence.
Comment: Line 70-ok so did you see these signs on the trail?
Answer: yes, specified in the sentence.
Comment: Line 74- checking the flag every 100 m is a bit too long. You could have potentially lost ticks.
There probably needs to be some discussion about how removal of ticks does/doesn't affect the tick population there
Answer: The transect was 100m, however ticks were checked every 20m; we specified this in the text. Dragging transects do not affect the tick population (see, as an example: Kjellander, P. L., Aronsson, M., Bergvall, U. A., Carrasco, J. L., Christensson, M., Lindgren, P. E., ... & Kjellander, P. (2021). Validating a common tick survey method: cloth-dragging and line transects. Experimental and Applied Acarology, 83(1), 131-146.). Additionally, being the same procedure repeated at all transects and the gradient between distances and not between time, this would not affect our results. The trends per sampling points (peaks and zero values) are correlated with the season rather than the sampling procedure.
Comment: Line 70- "broad" hiking trails. I did not see mention before of how wide the trails were and why it matters? That is actually an interesting question.
Answer: indeed the word was misleading and we removed it from the sentence. In its original meaning it wanted to highlight how the hiking trail is a “broad”, open, space with respect to the close vegetation on the side.
Comment: Line 135 Nymphs showed a decrease in number getting further from the trail (insert statistical significance) Same for the rest of the discussion. Insert statistical significance along with the descriptive statistics.
Answer: we indicated statistical significance.
Comment: Line 137 Carbon dioxide traps
Answer: corrected.
Comment: Line 138 139 decreased by
Answer: corrected.
Comment: Line 146 further away
Answer: corrected.
Comment: Line 173 "findings show" and "risk to"
Answer: corrected.
Reviewer 4 Report
Comments and Suggestions for Authors
This article presents a "non-evidence" in support of a hypothesis that ticks tend to gather in immediate vicinity of touristic paths elevating thus the risk for visitors. Scarcely two hundred specimens of all instars belonging to five tick species have been flagged along four paths in La Mandria Park to prove the hypothesis - a material that the authors closely scrutinized to extract the most of it. Nevertheless, Discussion concludes with a complain that - due to scarcity of ticks - the authors are unable to support the hypothesis with a full-fledged statistical evidence, just with visual impression of ticks-paths association on data plots (p.5, l.156 -p.6, l.158). Another weakness is the fact that no data on (micro)climatic conditions at the sampling sites were recorded. It could have aided in resolving whether ticks indeed "thrive where the host steps" or rather follow humidity patterns (note, for example, that soil along paths receives more rain that runs off their +/- compacted surface)..
Because the authors themselves present their results as falling short of rigorous scientific criteria, I was disposed to recommend article's rejection. However, it turned out that the statistics could be made more conclusive, and the article saved from rejection even without insisting on more robust data.
1/ Methodology is laconic but so much can be understood that the authors apply an unusual approach - they struggle to prove gradient in tick abundance by running multiple tests of contrast between adjacent zones (0 vs. 1m, 1 vs. 2m, 2 vs.4m) which, however, doesn't reflect the data in toto. The concept of how partial tests combine into an overall decision isn't explained and remains a mystery to me.
2/ To check performance of standard approaches, I picked data summarized in Fig.2, and tentatively calculated correlation tests, namely Kendall, Spearman (where rank is the independent variable), and Pearson (..distance in m). Assuming the alternative hypothesis that abundance of questing ticks decreases with path proximity - proportionally to ticks' chance to attach to a passing host and be carried away - I adopted the one-sided test variant (alternative="less"). Note that this assumption acts in favor of rejection of the null of independence...
3/ It showed that, under these conditions, gradient in the subset "nymphs", can be classified as "significant" even under more stringent conditions (Bonferroni correction), whereas that in the subset "Ixodes" can be rather epitheted as "loosely significant".
I recommend the authors to consult a professional statistician for a better advice.
Minor issues
P.2, l.53-4 there is no closed season even for gestation ?
P.2, l.59 delete "wild ungulates", pls
P.2, l.60, 66 is there really some free-ranging wolf population in the (fenced periurban) park?
P.2, l.64: sampling "points" 100 m long?? I propose the term "sampling sites" instead..
P.2, l.74-5: "A 1-square-meter white cloth was used for sampling and was checked at the end of each 100-meter transect" -isn't there a typo? If not it partially explains poor efficacy of collecting tick: dragging cloth should be inspected every ~10 metres, otherwise attached individuals, particularly adults, are quickly rubbed off again...
P.4, l.102-3: Dermacentor sp.
Table 2: rows 1, 3, and 4 do not sum to totals in col.#6
P.4, l.108-10 and throughout: round to one decimal, pls.
Fig. 2 (D): Ixodes ricinus
P.5, l.120: ditto
Table 3, header: ticks; row1: explain what "A","B", and "C" stand for, pls.
P.5, l.139: decreased by
P.5, l.146-9: "Conversely, small mammals, which are mainly hosts for larval stages, have a better access through dense vegetation and some species may also show avoidance behaviors towards trails, thus maintaining larvae abundance more consistent even at further distances" - illogical statement - to put it simply - small mammals sweep larvae to scatter nymphs - ergo, they don't "maintain larval abundance" but nymphal abundance...
P.5, l.152: Ixodes ricinus
P.6, l.162-167: I don't understand well what are the authors talking about - would they disentangle it a bit?
Author Response
We thank the reviewer for their comments and suggestions, which we incorporated into the manuscript, thus improving its overall quality.
We tested the reviewer's suggestion to improve statistics, but eventually, we decided to opt for a GAM (lines 127-136), which was more suitable to describe our trend. We updated our methods and results.
Minor issues
Comment: P.2, l.53-4 there is no closed season even for gestation?
Answer: from the culling plan for wild boar and red deer in the park (https://www.parchireali.it/parco.mandria/pagina.php?id=396), “Programmazione degli interventi. Si prevede che gli abbattimenti avvengano in tutti i mesi dell’anno ad esclusione di un periodo di 20 giorni prima delle date prefissate per i censimenti. Per il cinghiale: tutti i mesi dell’anno, gabbie attive 5 giorni su 7, mentre abbattimenti congiunti con cervo e daino 2 giorni su 7”. Translated: “Culling is scheduled to take place throughout the year, with the exception of a 20-day period before the set dates for population censuses. For wild boar: the same applies, with culling allowed all year round, traps active 5 days a week, and combined culling with deer and fallow deer on 2 days a week.”
Comment: P.2, l.59 delete "wild ungulates", pls
Answer: Corrected.
Comment: P.2, l.60, 66 is there really some free-ranging wolf population in the (fenced periurban) park?
Answer: yes, wolves were recorded by camera trap surveys, a reference was added.
Comment: P.2, l.64: sampling "points" 100 m long?? I propose the term "sampling sites" instead.
Answer: corrected across the whole paper.
Comment: P.2, l.74-5: "A 1-square-meter white cloth was used for sampling and was checked at the end of each 100-meter transect" -isn't there a typo? If not it partially explains poor efficacy of collecting tick: dragging cloth should be inspected every ~10 metres, otherwise attached individuals, particularly adults, are quickly rubbed off again...
Answer: transects were 100m length, but dragging cloth was checked every 20m. We specified this in the text.
Comment: P.4, l.102-3: Dermacentor sp.
Answer: corrected.
Comment: Table 2: rows 1, 3, and 4 do not sum to totals in col.#6
Answer: corrected.
Comment: P.4, l.108-10 and throughout: round to one decimal, pls.
Answer: corrected.
Comment: Fig. 2 (D): Ixodes ricinus
Answer: corrected
Comment: P.5, l.120: ditto
Answer: corrected.
Comment: Table 3, header: ticks; row1: explain what "A","B", and "C" stand for, pls.
Answer: sampling sites, added to the columns
Comment: P.5, l.139: decreased by
Answer: corrected
Comment: P.5, l.146-9: "Conversely, small mammals, which are mainly hosts for larval stages, have a better access through dense vegetation and some species may also show avoidance behaviors towards trails, thus maintaining larvae abundance more consistent even at further distances" - illogical statement - to put it simply - small mammals sweep larvae to scatter nymphs - ergo, they don't "maintain larval abundance" but nymphal abundance...
Answer: the sentence was corrected.
Comment: P.5, l.152: Ixodes ricinus
Answer: corrected.
Comment: P.6, l.162-167: I don't understand well what are the authors talking about - would they disentangle it a bit?
Answer: rephrased to make it clearer.
Reviewer 5 Report
Comments and Suggestions for Authors
The manuscript, entitled "Where the host steps, the tick thrives: description of abundance trends of ticks sampled at different distances from hiking trails," requires a number of improvements prior to future publication.
The text describes the prevalence of ticks in relation to human activity, specifically the proximity of ticks to footpaths. The idea of such research is not novel; similar studies were conducted previously, for example:
“Rasi, Tomas, et al. "Tick distribution along animal tracks: implication for preventative medicine." Annals of Agricultural and Environmental Medicine 25.2 (2018): 360-363.”
or
“Dobson, A. D., Taylor, J. L., & Randolph, S. E. (2011). Tick (Ixodes ricinus) abundance and seasonality at recreational sites in the UK: hazards in relation to fine-scale habitat types revealed by complementary sampling methods. Ticks and tick-borne diseases, 2(2), 67-74.”
Furthermore, the results are neither unexpected nor surprising.
The paper requires significant revisions. Primarily, the title should be amended to align with the formal and specialized nature of the research. Secondly, the authors must conduct a thorough and meticulous examination of the references list, supplementing it with a more comprehensive and up-to-date array of sources. Currently, the list comprises only 17 positions, which is insufficient for academic discourse.
In the introduction, the author makes mention of studies conducted in Poland but does not provide examples from the literature that could be considered accurate. Additionally, the authors discuss distance estimated by Ixodes ricinus ticks but do not include information about locomotor activity, which should be, in my opinion, included here.
Add at least the most common tick-borne pathogens (diseases) to this section.
Additional flaws:
L28 – The sentence utilizing the metaphor of humans as shields is not sufficiently clear and should be rewritten.
The Materials and Methods section could be divided into subsections, such as Study Sites or Statistical Analysis, to enhance clarity.
L 52 - It would be beneficial to include the highest and lowest temperatures, which are important in a country like Italy.
L84 - in my opinion the transects are not red.
Table 2 indicates that Dermacentor reticulatus was identified, whereas Table 1 lists Dermacentor spp. Can the Authors confirm whether the larva was correctly identified as D. reticulatus? I am reluctant to accept this conclusion without molecular analysis.
I am led to believe that the ticks were analysed by suitably qualified personnel. However, I would suggest that the addition of photographic evidence would serve to corroborate the results of the identification process.
L 115 – Plot
Table 3 requires further clarification regarding the meaning of the letters A, B, and C. The rationale behind their inclusion was not made evident.
It is unfortunate that no studies were conducted during the months of adult tick activity. Furthermore, the studies only demonstrate the activity of juvenile stages, which constrains the research's comprehensiveness and comparability with other studies. Additionally, the data provided does not encompass the full spectrum of tick abundance. It may be beneficial to consider modifying the title to reflect this.
It is also unexpected that the data set does not include at least temperature and humidity measurements taken at the time of tick collection. Maybe the Authors should add at least satellite data, e.g. tutiempo.
It is strongly recommended that the discussion section be entirely rewritten and that new information be added, as well as that the position be discussed with other studies and that new references be included. You can use references below:
Iijima, H., Watari, Y., Furukawa, T., & Okabe, K. (2022). Importance of host abundance and microhabitat in tick abundance. Journal of Medical Entomology, 59(6), 2110-2119.
Van Gestel, M., Verheyen, K., Matthysen, E., & Heylen, D. (2021). Danger on the track? Tick densities near recreation infrastructures in forests. Urban Forestry & Urban Greening, 59, 126994.
But check also:
Mols, B., Badji-Churchill, J. E., Cromsigt, J. P., Kuijper, D. P., & Smit, C. (2022). Recreation reduces tick density through fine-scale risk effects on deer space-use. Science of the Total Environment, 839, 156222.
Please verify the entire manuscript with regard to the use of Latin names in italics, as these are sometimes overlooked.
Author Response
We thank the reviewer for their comments and suggestion, which we incorporated in the manuscript thus improving the overall quality of it.
The manuscript, entitled "Where the host steps, the tick thrives: description of abundance trends of ticks sampled at different distances from hiking trails," requires a number of improvements prior to future publication.
Comment: The text describes the prevalence of ticks in relation to human activity, specifically the proximity of ticks to footpaths. The idea of such research is not novel; similar studies were conducted previously, for example:
“Rasi, Tomas, et al. "Tick distribution along animal tracks: implication for preventative medicine." Annals of Agricultural and Environmental Medicine 25.2 (2018): 360-363.”
or
“Dobson, A. D., Taylor, J. L., & Randolph, S. E. (2011). Tick (Ixodes ricinus) abundance and seasonality at recreational sites in the UK: hazards in relation to fine-scale habitat types revealed by complementary sampling methods. Ticks and tick-borne diseases, 2(2), 67-74.”
Furthermore, the results are neither unexpected nor surprising.
Answer: we have implemented the introduction and discussion with current work on acarological risk for people in parks and along trail, introducing also the suggested literature.
The paper requires significant revisions.
Comment: Primarily, the title should be amended to align with the formal and specialized nature of the research.
Answer: the title was amended following the reviewer’s suggestion.
Comment: Secondly, the authors must conduct a thorough and meticulous examination of the references list, supplementing it with a more comprehensive and up-to-date array of sources. Currently, the list comprises only 17 positions, which is insufficient for academic discourse.
Answer: Reference list was implemented
In the introduction, the author makes mention of studies conducted in Poland but does not provide examples from the literature that could be considered accurate. Additionally, the authors discuss distance estimated by Ixodes ricinus ticks but do not include information about locomotor activity, which should be, in my opinion, included here.
Answer: few literature is indeed provided on the very same topic, for I. ricinus in particular. However, we have expanded the introduction mentioning studies about factors influencing questing and locomotor activity, as well as dispersal studies on other species (lines 54-57).
Comment: Add at least the most common tick-borne pathogens (diseases) to this section.
Answer: we added three examples.
Additional flaws:
Comment: L28 – The sentence utilizing the metaphor of humans as shields is not sufficiently clear and should be rewritten.
Answer: the sentence was rephrased to improve clarity.
Comment: The Materials and Methods section could be divided into subsections, such as Study Sites or Statistical Analysis, to enhance clarity.
Answer: Materials and methods were divided into subsections.
Comment: L 52 - It would be beneficial to include the highest and lowest temperatures, which are important in a country like Italy.
Answer: we added the minimum and maximum temperature.
Comment: L84 - in my opinion the transects are not red.
Answer: caption was corrected.
Comment: Table 2 indicates that Dermacentor reticulatus was identified, whereas Table 1 lists Dermacentor spp. Can the Authors confirm whether the larva was correctly identified as D. reticulatus? I am reluctant to accept this conclusion without molecular analysis.
Answer: indeed, as molecular analysis was not possible, we corrected to Dermacentor sp.
Comment: I am led to believe that the ticks were analysed by suitably qualified personnel. However, I would suggest that the addition of photographic evidence would serve to corroborate the results of the identification process.
Answer: Yes, identification was provided by qualified personnel:
Battisti, E., Zanet, S., Boraso, F., Minniti, D., Giacometti, M., Duscher, G. G., & Ferroglio, E. (2019). Survey on tick-borne pathogens in ticks removed from humans in Northwestern Italy. Veterinary Parasitology: Regional Studies and Reports, 18, 100352.
Maurelli, M. P., Pepe, P., Colombo, L., Armstrong, R., Battisti, E., Morgoglione, M. E., ... & Zanet, S. (2018). A national survey of Ixodidae ticks on privately owned dogs in Italy. Parasites & Vectors, 11, 1-10.
If needed, we can add a supplementary file with some picutres, however it would not be very useful to assess the quality of ticks identification, as our instruments are not intended to take pictures of publication quality.
Comment: L 115 – Plot
Answer: corrected.
Comment: Table 3 requires further clarification regarding the meaning of the letters A, B, and C. The rationale behind their inclusion was not made evident.
Answer: we corrected the table heading to improve clarity
Comment: It is unfortunate that no studies were conducted during the months of adult tick activity. Furthermore, the studies only demonstrate the activity of juvenile stages, which constrains the research's comprehensiveness and comparability with other studies. Additionally, the data provided does not encompass the full spectrum of tick abundance. It may be beneficial to consider modifying the title to reflect this.
Answer: Most studies have been focusing on nymphal abundance, as the developmental stage representing the higher zoonotic risk (i.e. Millet et al., Assessment of the Exposure of People to Questing Ticks Carrying Agents of Zoonoses in Aosta Valley, Italy or Van Gestel et al., Danger on the track? Tick densities near recreation infrastructures in forests). However, it is true that few adults have been collected in the study and we changed the title accordingly.
Comment: It is also unexpected that the data set does not include at least temperature and humidity measurements taken at the time of tick collection. Maybe the Authors should add at least satellite data, e.g. tutiempo.
Answer: we had temperature and humidity data recorded by sensors in the proximity of the sampled sites, we implemented this data in the analysis as Saturation Deficit.
Comment: It is strongly recommended that the discussion section be entirely rewritten and that new information be added, as well as that the position be discussed with other studies and that new references be included. You can use references below:
Iijima, H., Watari, Y., Furukawa, T., & Okabe, K. (2022). Importance of host abundance and microhabitat in tick abundance. Journal of Medical Entomology, 59(6), 2110-2119.
Van Gestel, M., Verheyen, K., Matthysen, E., & Heylen, D. (2021). Danger on the track? Tick densities near recreation infrastructures in forests. Urban Forestry & Urban Greening, 59, 126994.
But check also:
Mols, B., Badji-Churchill, J. E., Cromsigt, J. P., Kuijper, D. P., & Smit, C. (2022). Recreation reduces tick density through fine-scale risk effects on deer space-use. Science of the Total Environment, 839, 156222.
Answer: discussion was extended and implemented with further citations, including the ones suggested.
Comment: Please verify the entire manuscript with regard to the use of Latin names in italics, as these are sometimes overlooked.
Answer: Latin names were corrected
Round 2
Reviewer 2 Report
Comments and Suggestions for Authors
I checked the reviews against the revised manuscript and believe that the authors have done a commendable job.
Comments on the Quality of English LanguageI checked the reviews against the revised manuscript and believe that the authors have done a commendable job. They have adequately addressed the primary questions raised in the peer reviews. I believe the paper warrants publication in Veterinary Sciences, and will serve as an important citation and model for future study in this increasingly important research area of human health and zoonoses.
Author Response
On behalf of all authors, I thank the final assessment of the reviewer
Reviewer 3 Report
Comments and Suggestions for Authors
This is much better. There are still a few things that need correcting
Line 54. Change to: " When not on the host", ticks are..
Graphs before line 155 would be much more clear if you inserted collection dates into the x axis. Sampling site 5? , 10?, 11? Are you saying you had other sites were ticks weren't collected? I'm sorry, as I think I missed that information. Also why did you not include in the graph? Too many sites? If this was just a number then I suggest changing to site A, B, C to reduce confusion.
Author Response
Comment: Line 54. Change to: " When not on the host", ticks are..
Answer: corrected.
Comment: Graphs before line 155 would be much more clear if you inserted collection dates into the x axis. Sampling site 5? , 10?, 11? Are you saying you had other sites were ticks weren't collected? I'm sorry, as I think I missed that information. Also why did you not include in the graph? Too many sites? If this was just a number then I suggest changing to site A, B, C to reduce confusion.
Answer: the graph was corrected according to suggestions. There was indeed a typo in naming the sampling sites, wich are always the same (A, B, C). Everything was corrected and implemented as suggested.
Reviewer 4 Report
Comments and Suggestions for Authors
I acknowledge that the manuscript has been improved appreciably. Still, there persist some issues needing correction and thwarting in recommending this - otherwise elaborate - contribution for publication
P.1, l.3: ..trails in a natural park..
P.1, l.28-9: ..decrease with distance..; delete "increased"
P.1, l.39: suggestion: substitute "herbivores" for "prey species"
P.1. l.41: predators' activity (3), whereas
P.1, l.42: such a behavior
P.2, l.43: avoidance of human tracks
P.2, l.65: lower cases in: "Babesiosis, Tick-Borne Encephalitis"
P.2, l.65-6 : " it may inform infrastructure design and guide wildlife and vegetation management practices in peri-urban parks" - a phrase, reword it, pls (what about noting that results can serve a guidance for placing tick warning signs - providing this measure is practiced in the park at all, of course.. ??)
P.2, l.67: ..determine the gradient in ticks’ distribution... ?
P.2, l.78 and throughout: round to one decimal, pls, - note it is nonsensical to pretend precision exceeding the measurement error
P.3, i.104: " and was checked at the end of each 100-meter transect" - it should have been corrected already !!! -> ..and was checked every 20 m... !!!!
P.3, l.107: " All points were sampled" ?? -> all sites were sampled !!
P.3, l.111: "In the proximity of the sampling sites, we placed a sensor..." - be clear, pls, as to whether each site had assigned its own logger or a single common logger was placed somewhere in between the sites
P.3, l.112: "Data Logger RS PRO" - the model number could be specified...
P.3, l.122: at different distances
P.3, l.123: definition of "average abundance of tick" is missing - insert, say, "hereinafter defined as the number of questing ticks captured per square metre", or so..
P.3, l.124: disregarding adults ?
P.4, l.127-9: " We assessed the statistical significance of the changes in abundance with a Mann-Whitney test among the three pairs of distances (0m to 1m, 1m to 2m, 2m to 4m)" - Why on Earth the authors still stick to this procedure if it showed completely nothing? By this method they fail to show any significant difference among the transect pairs (p.6,l169-74), to arrive next, based on GAM, at an opposite conclusion (p.7, l.177-p.8,l.194). This section should be revised to make thinks consistent.
P.5, l.156-7: " Fluctuation of tick abundance across the 9 repetitions in the three sampling sites: repetition 1 is on 3rd of May, while repetition 9 is on 24th of August " - confusing and complicated - "repetitions" should be translated into dates and plotted as a plain seasonal diagram.
P.7, l.179-80: upper case in "Poisson"
P.7, Table 4: round to a reasonable number of decimals, pls.
P.7, l.188 0.5
P.8, Fig.4: As explaining in their cover letter, inspired by a sound idea of employing additional covariates, the authors 'contempt' of simple (e.g. correlation) techniques of gradient detection and opted for GAM. In comparison to simple tests (that just barely reached significance with the limited authors' data in my trials), GAM is rather data-hungry. In this context, I'm astonished at incredibly narrow fiducial bands presented in Fig.4. I urge the authors to check, and double-check.., whether it is correct...?
P.8, l.205: than those reported
P.8, l.216: sampling site
Author Response
Thanking the reviewer for the new correction, hereby our answers:
Comment: P.1, l.3: ..trails in a natural park..
Answer: corrected
Comment: P.1, l.28-9: ..decrease with distance..; delete "increased"
Answer: corrected
Comment: P.1, l.39: suggestion: substitute "herbivores" for "prey species"
Answer: we believe that herbivores is not an exhaustive term to encompass prey species
Comment: P.1. l.41: predators' activity (3), whereas
Answer: corrected
Comment: P.1, l.42: such a behavior
Answer: corrected
Comment: P.2, l.43: avoidance of human tracks
Ansewer: corrected
Comment: P.2, l.65: lower cases in: "Babesiosis, Tick-Borne Encephalitis"
Answer: corrected
Comment: P.2, l.65-6 : " it may inform infrastructure design and guide wildlife and vegetation management practices in peri-urban parks" - a phrase, reword it, pls (what about noting that results can serve a guidance for placing tick warning signs - providing this measure is practiced in the park at all, of course.. ??)
Answer: sentence was rephrased giving a more comprehensive meaning
Comment: P.2, l.67: ..determine the gradient in ticks’ distribution... ?
Answer: corrected
Comment: P.2, l.78 and throughout: round to one decimal, pls, - note it is nonsensical to pretend precision exceeding the measurement error
Answer: corrected
Comment: P.3, i.104: " and was checked at the end of each 100-meter transect" - it should have been corrected already !!! -> ..and was checked every 20 m... !!!!
Answer: this sentence was a refuse, we corrected the text.
Comment: P.3, l.107: " All points were sampled" ?? -> all sites were sampled !!
Answer: corrected
Comment: P.3, l.111: "In the proximity of the sampling sites, we placed a sensor..." - be clear, pls, as to whether each site had assigned its own logger or a single common logger was placed somewhere in between the sites
Answer: each sampling site, corrected in the text
Comment: P.3, l.112: "Data Logger RS PRO" - the model number could be specified...
Answer: the model was added.
Comment: P.3, l.122: at different distances
Answer: corrected.
Comment: P.3, l.123: definition of "average abundance of tick" is missing - insert, say, "hereinafter defined as the number of questing ticks captured per square metre", or so..
Answer: we specified average abundance of ticks per transect
Comment: P.3, l.124: disregarding adults ?
Answer: corrected
Comment: P.4, l.127-9: " We assessed the statistical significance of the changes in abundance with a Mann-Whitney test among the three pairs of distances (0m to 1m, 1m to 2m, 2m to 4m)" - Why on Earth the authors still stick to this procedure if it showed completely nothing? By this method they fail to show any significant difference among the transect pairs (p.6,l169-74), to arrive next, based on GAM, at an opposite conclusion (p.7, l.177-p.8,l.194). This section should be revised to make thinks consistent.
Answer: Indeed, we have changed the presentation of results, eliminating the Mann-Whitney test that did not provide any statistical significance and leaving only descriptive statistics and GAM results.
Comment: P.5, l.156-7: " Fluctuation of tick abundance across the 9 repetitions in the three sampling sites: repetition 1 is on 3rd of May, while repetition 9 is on 24th of August " - confusing and complicated - "repetitions" should be translated into dates and plotted as a plain seasonal diagram.
Answer: graph was corrected, as well as the caption
Comment: P.7, l.179-80: upper case in "Poisson"
Answer: corrected
Comment: P.7, Table 4: round to a reasonable number of decimals, pls.
Answer: corrected
Comment: P.7, l.188 0.5
Answeer: corrected
Comment: P.8, Fig.4: As explaining in their cover letter, inspired by a sound idea of employing additional covariates, the authors 'contempt' of simple (e.g. correlation) techniques of gradient detection and opted for GAM. In comparison to simple tests (that just barely reached significance with the limited authors' data in my trials), GAM is rather data-hungry. In this context, I'm astonished at incredibly narrow fiducial bands presented in Fig.4. I urge the authors to check, and double-check.., whether it is correct...?
Answer: The outputs of our GAM are indeed constrained by the low number of collected ticks, which is why the deviance in the data is not perfectly explained by our variables (from 22 to 36%), which is already discussed in the text. The higher k value for the smooth term, relative to the distance levels, was intentionally selected to more effectively capture the trends observed in the descriptive statistics, resulting in narrow fiducial bands.
Comment: P.8, l.205: than those reported
Answer: corrected
Comment: P.8, l.216: sampling site
Answer: corrected
Reviewer 5 Report
Comments and Suggestions for Authors
I am satisfied with the revisions proposed by the Authors. I believe that the manuscript, in its present form, meets the standards required for publication.
Author Response
On behalf of all authors, we thank the final recommendation of the reviewer